# GATED FAST WEIGHTS FOR ASSOCIATIVE RETRIEVAL

## ABSTRACT

We improve previous end-to-end differentiable neural networks (NNs) with fast weight memories. A gate mechanism updates fast weights at every time step of a sequence through two separate outer-product-based matrices generated by slow parts of the net. The system is trained on a complex sequence to sequence variation of the Associative Retrieval Problem with roughly 70 times more temporal memory (i.e. time-varying variables) than similar-sized standard recurrent NNs (RNNs). In terms of accuracy and number of parameters, our architecture outperforms a variety of RNNs, including Long Short-Term Memory, Hypernetworks, and related fast weight architectures.

## 1 INTRODUCTION

Recurrent Neural Networks (RNNs) are general parallel-sequential computers that can implement algorithms which map input sequences to output sequences. One variation of it, the *Long Short-Term Memory* (LSTM), has achieved great success on a wide variety of Machine Learning tasks such as natural language translation, image caption generation, and speech recognition among others Hochreiter & Schmidhuber (1997); Gers et al. (2000); Graves et al. (2009). In practical applications, most RNNs are actually LSTM networks now used billions of times per day for automatic translation Wu et al. (2016), speech recognition Sak et al., and many other tasks Sak et al.; Schmidhuber (2015).

However, plain RNNs but also LSTMs are known to have difficulty in performing memorization, like e.g. a simple copying task of outputting the same sequence as the input sequence Zaremba & Sutskever (2014). But also other more high-level cognitive tasks have been shown to be difficult to master Danihelka et al. (2016).

In this work, we explore a generalization of the Associative Retrieval problem. We follow a similar style as in Danihelka et al. (2016) but turned the task into a general sequence to sequence problem and also substantially increased its complexity. The underlying mechanism is essentially a dictionary with a certain number of key-value pairs which is controlled using a simple syntax of storage and query tokens.

In order to overcome the limitation of current RNNs on this task, we propose a fast weight architecture that is able to learn and generalize using much fewer parameters. Our architecture consists of the two networks $s$ and $f$ which both operate on the input sequence in parallel. The small network $f$ predicts the targets while the big network $s$ generates on-the-fly weight-updates for $f$. The big network $s$ is called the slow network because its weights change only after every mini-batch according to the gradient-based learning algorithm. $f$, on the other hand, is called the fast network because its weights can change after every time step.

## 2 RELATED WORK

Our work is heavily influenced by early work in Neural Networks. Like many ideas at the time, the idea of fast-changing weights emerged out of biological evidence and the efforts of storing activation patterns in the weights of an associative network. Networks with non-differentiable fast weights or "dynamic links" have been published since 1981 von der Malsburg (1981); Feldman (1982); Hinton & Plaut (1987). Subsequent work showed that a slow network can use gradient descent learning to control fast weights of a separate network in end-to-end differentiable fashion Schmidhuber (1992). A more recent work on fast weights, which provides a good overview of the physiological facts, is

the work by Ba et al. (2016a). In their work they aptly describe their own fast weight architecture as an attention mechanism which effectively increases the flexibility of the learned program, making it more adaptive through time and able to store temporally recent information but, by design, it is unable to store long-term knowledge using that same mechanism. Their decay of the modulation on top of the slow weights that long-term knowledge has to be learned using a learning algorithm like gradient descent and can't be learned using fast weights.

Our approach follows more from Schmidhuber (1992) which frames the fast weights idea as a means of program generation i.e. using the first network to produce context dependent weight updates for a second network. We also refer to the weights of a network as its *program*. An adaptive program, such as the one of $f$, and its benefits can be described through the concept of the ratio of time-varying variables Schmidhuber (1993a). Take as an example a plain RNN. The program is its recurrent weight matrix. Once trained, the number of time-varying variables in the computational graph is limited to the state variables and increasing the size of the recurrent weight matrix quadratically increases the number of weight variables available for the implementation of the during inference static program but only linearly increase the number of adaptive state variables. Fast weight architectures tend to relax this ratio allowing the program to also change through time in order to adapt to the context-specific needs.

An adjacent approach to the relaxation of the ratio of time-varying variables is the Hypernetwork by Ha et al. (2016). Their initial idea consisted of an adaptive interpolation of several Long Short-Term Memory (LSTM) cells but then lead to a novel approximation which still allows for an adaptive program. Their sequential Hypernetwork was applied to language modelling and outperformed the LSTM on several datasets.

## 3 METHOD

As introduced, our fast weight architecture consists of a big network $s$ and a small network $f$. For simplicity we chose both networks to be RNNs but they could be replaced by any other recurrent architecture, such as a LSTM. While the weights of $s$ change after every batch, the weights of $f$ change after every time step, hence the fast/slow dichotomy. The following formulas are for a single sample or a batch size of 1 and uppercase letters refer to matrices while lower case letters refer to vectors. The biases are omitted for simplicity.

Both, the slow network $s(x_t, h_t^S)$ and the time-varying fast network $f_t(x_t, h_t^F)$ use the same input embedding and the same output projection as it is common practice for RNNs. In this paper, we use in both cases a 2-layer transition RNN. Recall the basic RNN formulation,

$$h_{t+1} = \phi(W[h_t; x_t]) \tag{1}$$

where $h$ is the hidden state, $\phi$ is a non-linear activation function such as $\tanh$, $x_t$ is the current input from the input sequence $X = (x_1, x_2, ..., x_T)$, and the weights $W \in \mathbb{R}^{m \times n}$ are fixed parameters.

Analogously, we define our fast network $f_t(x_t, h_t^F)$:

$$h_{t+1}^F = \mathcal{LN}(\tanh(F_t^{(2)} \mathcal{LN}(\tanh(F_t^{(1)}[h_t^F; x_t])))) \tag{2}$$

Where $h^F$ is the hidden state of the fast network, $\mathcal{LN}(.)$ refers to layer normalization (LN) as introduced by Ba et al. (2016b), and $F^{(1)} \in \mathbb{R}^{m \times n}, F^{(2)} \in \mathbb{R}^{m \times m}$ are parameters which are not fixed and can change through time.

Similarly, we define the the slow network $s(x_t, h_t^S)$:

$$[z_t^S; \Delta_t^{(1)}; \Delta_t^{(2)}] = S^{(2)} \tanh(S^{(1)}[h_t^S; x_t]) \tag{3}$$

$$h_{t+1}^S = \tanh(z_t^S) \tag{4}$$

Where $h^S$ is the hidden state of the slow network, $S^{(1)} \in \mathbb{R}^{p \times o}, S^{(2)} \in \mathbb{R}^{p \times p}$ are fixed parameters, and $\Delta_t^{(1)} \in \mathbb{R}^{2(n+m)}$ and $\Delta_t^{(2)} \in \mathbb{R}^{(4m)}$ are the update representations for $F_{t+1}^{(1)}$ and $F_{t+1}^{(2)}$

respectively. Every $F_{t+1}$ is then calculated as follows:

$$[\alpha_t; \beta_t; \gamma_t; \delta_t] = \Delta_t \tag{5}$$

$$H_t = \tanh(\alpha_t)\tanh(\beta_t)^T \tag{6}$$

$$T_t = \sigma(\gamma_t)\sigma(\delta_t)^T \tag{7}$$

$$F_{t+1} = T_t \odot H_t + (1 - T_t) \odot F_t \tag{8}$$

Where $H$ and $T$ are outer products to generate weight matricies in an Hebb-like manner as introduced by Schmidhuber (1992) and have the same dimensionality as the respective $F$, $\odot$ is the element-wise product, 1 denotes a matrix of ones.

We make use of LN in the fast network because it has shown to stabilize learning especially in the beginning of the training but also slightly improved validation set performance. Because of the generated nature of the fast weights, we expected them to be far from good initial values when training begins. We use LN to overcome this problem. We tested our architecture without LN, with pre-activation LN, with post-activation LN, and with pre and post-activation LN and found that using a post-activation LN always works best no matter if we normalize pre-activation.

In equation 8 we use a gating matrix to blend the current fast matrix with a matrix update. This is essentially equal to the gating mechanism for activations in a highway network Srivastava et al. (2015). We also evaluated other multiplicative and additive interactions using more than one matrix but achieved the best results using this specific update mechanism.

Furthermore, we'd like to point out that the slow network $s$ is generating weights for the next time step. We do this to prevent $s$ from fully learning a prediction on its own which bypasses $f$. We didn't experiment with update delays greater than one step.

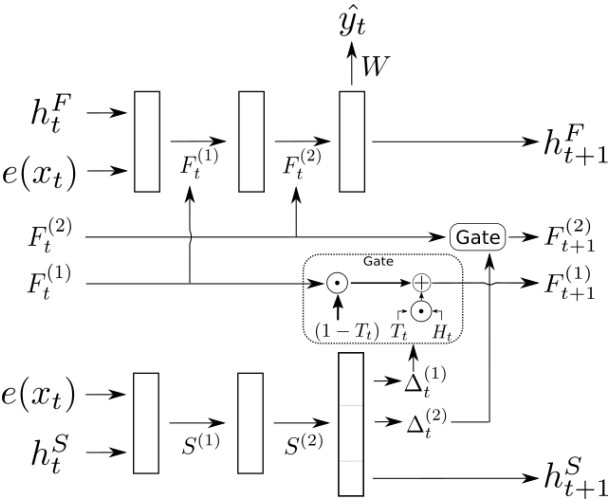

Figure 1: An informal pictogram which visualises the fast and slow network of our architecture at time step $t$. $e(x_t)$ refers to the embedding of $x_t$ and biases, activation functions, layernorm, and outer-products are not displayed.

## 4 EXPERIMENTS

We created a challenging sequence to sequence task based on the Associative Retrieval problem. Our version employs of a simple syntax of storage and query tokens. Storage tokens are key-value pairs while query tokens come with only a key to which the network must output the respective value according to a previously seen storage token. All tokens are part of the inputs sequence while the generated answers to the query tokens are part of the output sequence. Whenever there is no query to respond to the network is supposed to output some default value which in our case is the empty space character. This means that non-trivial problem-relevant predictions are rather sparse.

The keys are 2 to 4 characters long while the respective values are a single character. All characters are uniformly sampled with replacement from a set of 8 possible ASCII characters (a to h). Each query token has to be a valid key which the network must have seen after the previous query token. Preceding every query token there are between 1 and 10 storage tokens. We concatenated all query tokens and their respective storage tokens into one large sequence and use truncated Backpropagation Through Time (truncated BPTT) to train the network. The following is an example with only 2 queries with quotes to show the beginning and end of both sequences.

```
x: "S(hgb,c),S(ceaf,e),S(df,g),S(hac,b),Q(ceaf)e.S(hf,h),S(cc,d),Q(cc)d."
y: "                                      e                            d  "
```

We generate and concatenate 100'000 queries for the training set and 5'000 queries for the test and validation set. This results in a single training sequence of roughly 5.7 million characters and a test and validation sequence of roughly 288'000 characters each. Instead of measuring the accuracy of all predictions we use the *partial accuracy* which is the percentage of the correct predictions of non-space characters. This is because all models learn very quickly to predict spaces at every step which very quickly yields a high accuracy without actually having learned much. However, the cost and respective gradient are computed using all predictions. For all models and experiments, we used the Nesterov accelerated Adam (Nadam) by Dozat (2016). We observed in our experiments that Adam achieves similar performance but tends to converge slower than Nadam.

**The Model**   Our final architecture uses an embedding size of 15. The fast network is defined such that $h^F \in \mathbb{R}^{40}, F^{(1)} \in \mathbb{R}^{55 \times 40}, F^{(2)} \in \mathbb{R}^{40 \times 40}$ and the slow network such that $h^S \in \mathbb{R}^{40}, S^{(1)} \in \mathbb{R}^{55 \times 100}, S^{(2)} \in \mathbb{R}^{100 \times 394}$ both using a shared embedding of $\mathbb{R}^{15 \times 15}$ and a shared output projection $W \in \mathbb{R}^{40 \times 15}$ summing up to 46'234 trainable parameters (bias not listed for simplicity). We aimed for a small and very context specific fast network and a slow network big enough to support such a fast-changing fast network while using as few weights as possible. Different configurations are possible and a larger $s$ seems to help the model to converge faster but doesn't help performance. Preliminary experiments with different RNNs, like an RHN or LSTM as the fast and slow network, also showed promising results but were not part of this work.

**Results**   We provide our best experimental results for other architectures for which we performed a hyper parameter search using a mix of grid and random search over each architecture's individual parameters. We compare our architecture to the LSTM, fast weights as a means of attention to the recent past (AttentionFW) by Ba et al. (2016a), and HyperNetworks by Ha et al. (2016). We also performed experiments on the feed-forward fast weights and recurrent fast weights as introduced by Schmidhuber (1992; 1993b) but we were unable to get them to predict anything else than the trivial output on this task. We trained all models using a sequence length of 32 and a batch size of 256. We didn't perform learning rate decay or similar strategies but included the learning rate in our hyper parameter search. In the end, a learning rate of 0.002 achieved over all architectures the best results and was in a second phase fixed for all models to allow for a comparison of convergence qualities.

Our model achieves the highest partial accuracy and the lowest partial bits per character (BPC) while using 83 times fewer parameters than the next best model. Again, "partial" refers only to the non-space prediction targets.

Table 1: The test set results of the best models.

| Model | Total Accuracy | Partial Accuracy | Total BPC | Partial BPC | Parameters |
|---|---|---|---|---|---|
| AttentionFW | 0.9922 | 0.5323 | 0.0274 | 0.0063 | 100'140 |
| LSTM | 0.9936 | 0.6252 | 0.0267 | 0.0061 | 1'487'640 |
| Hypernetwork | 0.9963 | 0.7804 | **0.0137** | 0.0031 | 3'848'215 |
| ours | **0.9979** | **0.9522** | 0.0149 | **0.0016** | **46'234** |

## 5   DISCUSSION

It has been shown before how generalizing a memory mechanism, such as required in this task, is difficult for vanilla RNNs to learn. Several previous works focused on integrating differentiable

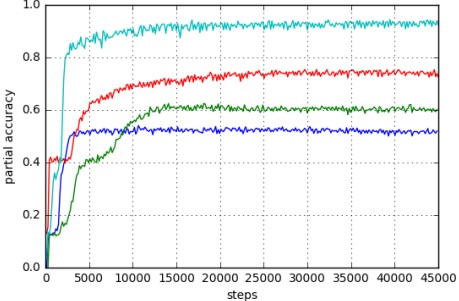 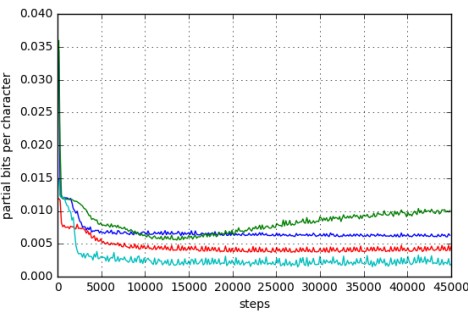

Figure 2: The left figure represents the accuracy of non-trivial targets and the right figure the respective bits per character. These are the validation set results of the best models of the four examined architectures due to our hyper parameter search. Green is the LSTM, dark blue is the fast weights architecture as attention to the recent past, red is the hypernetwork, and cyan is our novel fast weight architecture.

computer-like memory into the graph structure of the architecture such that the model wouldn't need to learn the mechanism itself but mainly how to use it. Examples of such are differentiable stacks by Das et al. (1992); Mozer & Das (1993), but also related storage types like those in LSTM-controlled Neural Turing Machines Graves et al. (2014) or memory nets Weston et al. (2014).

A basic argument against a memory approach inspired by the Turing Machine or the von Neumann architecture is its biological plausibility, as well as, the fact that we know how the human memory system often doesn't really behave as computer memory does. It is generally known to be much more nuanced and forcing an architecture to include a strong and possibly misleading bias would certainly limit its ability to learn and generalize to a more effective mechanism. We think that learning high-level cognitive functions (i.e. high-level programs implemented under the constraints of some artificial neural substrate) is difficult and find the idea to search and reverse engineer every human capability necessary for intelligence in order to engineer it into an architecture to be undesirable. Instead, we favour an approach which focuses on improving the capabilities of the artificial neural substrate which allows for the emergence of higher-level functions through training. We think fast weights are such a component from which many models could benefit.

**Limitations** Fast weights seem to have a positive effect when they are incorporated into an architecture but we experienced at least two practical limitations. While the calculation of the gradient through these fast weight dynamics remains rather cheap, the number of values to be stored in the backward pass encompasses now all time-varying variables (i.e. all fast weights) at each relevant time step. This quadratically increases the memory consumption compared to a similar sized RNN. At the moment, these memory limitations are the main reason why such fast weight networks remain rather small compared to state-of-the-art RNNs one some popular application like e.g. neural machine translation. Another noteworthy limitation is the wall-time necessary for computing a more complex architecture. Reshaping tensors and other simple operations result in a significant increase of wall-time.

Table 2: Wall-time for 5k steps on a P100 Nvidia GPU with roughly 50k trainable parameters per architecture.

| Model | Wall time (min) | Relative factor |
|---|---|---|
| AttentionFW | 22 | 1.6 |
| LSTM | 14 | 1.0 |
| Hypernetwork | 20 | 1.4 |
| ours | 25 | 1.8 |

**Time-Varying Variables** Standard RNNs have a very small ratio between time-varying variables and gradient-descent-learnable parameters. Time-varying variables allow the model to store information through time. In a standard RNN, this is only accomplished through its state vector. However, over 20 years ago it was pointed out that an RNN can also use additional, soft, end-to-end differentiable attention mechanisms to learn to control its own internal spotlights of attention Schmidhuber (1993a) to quickly associate self-defined patterns through fast weights (on connections between certain units) that can quickly and dramatically change from one time step to the next. This approach can essentially increase the number of time-varying variables massively while keeping the model relatively small.

We improved the update mechanism through which the slow network learns to write into its fast weight memory. This allows us to construct a model with a small but memory expensive fast network in addition to the standard slow network. However, the fast weights are not just passive memory like the state but are more like active memory in the sense of a context-specific computation. We force the model to use this active memory at every step to predict the current output by delaying the weight updates from slow network by one step.

Consider the model introduced in the previous section. While the slow network is technically bigger, it contains only 40 time-varying variables, namely the state vector $h^S$. The fast network is much smaller but has 3840 time-varying variables ($h^F$, $F^{(1)}$, and $F^{(2)}$). Increasing the total number of time-varying variables significantly.

## 6 CONCLUSION

In this paper, we introduce a complex sequence to sequence variation of the Associative Retrieval problem. In that problem, the model has to learn how to store a number of associations from the input sequence, retrieve them if necessary, and forget them to learn new associations. We use a standard RNN to generate weight updates for a fast weight RNN. This allows our model to store temporal information not only in the state of either RNN but also in the weights of the fast weight RNN. Our contribution is a new way of updating the weight matrices of the fast weight RNN where we use a gate and two generated matrices instead of one. Without our contribution the model has never shown to be able to learn any non-trivial predictions. We compare it with other architectures on this general task and show how it outperforms them in convergence, accuracy, and number of parameters.

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
