# OpenReview forum: "GATED FAST WEIGHTS FOR ASSOCIATIVE RETRIEVAL"
_ICLR.cc/2018/Conference — Reject_

### Official Review · AnonReviewer2 · 2017-11-26
**Interesting evolution of the fast weights idea lacking proper evaluation and strong baselines**

**Rating:** 3
**Confidence:** 5

**Review:**

The authors present an evolution of the idea of fast weights: training a double recurrent neural network, one "slow" trained as usual and one "fast" that gets updated in every time-step based on the slow network. The authors generalize this idea in a nice  way and present results on 1 experiment. On the positive side, the paper is clearly written and while the fast-weights are not new, the details of the presented method are original. On the negative side, the experimental results are presented on only 1 experiment with a data-set and task made up by the authors. The results are good but the improvements are not too large, and they are measured over weak baselines implemented by the authors. For a convincing result, one would require an evaluation on a number of tasks, including long-studied ones like language modeling, and comparison to stronger related models, such as the Neural Turing Machine or the Transformer (from "Attention is All You Need"). Without comparison to stronger baselines and with results only on 1 task constructed by the authors, we have to recommend rejection.

---

### Official Review · AnonReviewer3 · 2017-11-27
**Results seem good but only a single task. Hard to extract the most important contributions.**

**Rating:** 5
**Confidence:** 4

**Review:**

Summary
The paper proposes a neural network architecture for associative retrieval based on fast weights with context-dependent gated updates. The architecture consists of a ‘slow’ network which provides weight updates for the ‘fast’ network which outputs the predictions of the system. The experiments show that the architecture outperforms a couple of related models on an associative retrieval problem.

Quality
The authors evaluate their architecture on an associative retrieval task which is similar to the variable assignment task used in Danihelka et al. (2016). The difference with the original task seems to be that the network is also trained to predict a ‘blank’ symbol which indicates that no prediction has been made. While this task is artificial, it does make sense in the context of what the authors want to show. The fact that the authors compare their results with three sensible baselines and perform some form of hyper-parameter search for all of the models, adds to the quality of the experiment. It is somewhat unfortunate that the paper doesn’t give more detail about the precise hyper-parameters involved and that there is no comparison with the associative LSTM from Danihelka et al. Did these hyper-parameters also include the sizes of the models? Otherwise it’s not very clear to me why the numbers of parameters are so much higher for the baseline models. While I think that this experiment is well done, it is unfortunate that it is the only experiment the authors carried out and the paper would be more impactful if there would have been results for a wider variety of tasks. It is commendable that the authors also discuss the memory requirements and increased wall clock time of the model.

Clarity
I found the paper hard to read at times and it is often not very clear what the most important differences are between the proposed methods and earlier ones in the literature. I’m not saying those differences aren’t there, but the paper simply didn’t emphasize them very well and I had to reread the paper from Ba et al. (2016) to get the full picture.

Originality/Significance
While the architecture is new, it is based on a combination of previous ideas about fast weights, hypernetworks and activation gating and I’d say that the novelty of the approach is average. The architecture does seem to work well on the associative retrieval task, but it is not clear yet if this will also be true for other types of tasks. Until that has been shown, the impact of this paper seems somewhat limited to me.

Pros
Experiments seem well done.
Good baselines.
Good results.

Cons
Hard to extract the most important changes from the text.
Only a single synthetic task is reported.

---

### Official Review · AnonReviewer1 · 2017-11-28

**Rating:** 4
**Confidence:** 4

**Review:**

The paper proposed an extension to the fast weights from Ba et al. to include additional gating units for changing the fast weights learning rate adaptively. The authors empirically demonstrated the gated fast weights outperforms other baseline methods on the associative retrieval task.

Comment:

- I found the paper very hard to follow. The authors could improve the clarity of the paper greatly by listing their contribution clearly for readers to digest. The authors should emphasize the first half of the method section are from existing works and should go into a separate background section.

- Overall, the only contribution of the paper seems to be the modification to Ba et al. is the Eq. (8). The authors have only evaluated the method on a synthetic associative retrieval task. Without additional experiments on other datasets, it is hard for the reader to draw any meaningful conclusion about the proposed method in general.

---

### Decision · Program_Chairs · 2018-01-29
**ICLR 2018 Conference Acceptance Decision**

**Decision:**

Reject

**Comment:**

The reviewers agree that while the presented result looks interesting, it is but one result. Further, one of the reviewer finds this to be a weak comparison as well.
The novelty of the approach over the paper by Ba et. al. also is in question -- good results on multiple tasks might have made it worth exploring, but the authors did not establish this to be the case convincingly.